# Longitudinal NMR-Based Metabolomics Study Reveals How Hospitalized COVID-19 Patients Recover: Evidence of Dyslipidemia and Energy Metabolism Dysregulation

**DOI:** 10.3390/ijms25031523

**Published:** 2024-01-26

**Authors:** Laura Ansone, Vita Rovite, Monta Brīvība, Lauma Jagare, Līva Pelcmane, Daniella Borisova, Anne Thews, Roland Leiminger, Jānis Kloviņš

**Affiliations:** 1Latvian Biomedical Research and Study Centre, LV-1067 Riga, Latvia; vita.rovite@biomed.lu.lv (V.R.); monta.briviba@biomed.lu.lv (M.B.); lauma.jagare@biomed.lu.lv (L.J.); liva.pelcmane@biomed.lu.lv (L.P.); daniella.borisova@biomed.lu.lv (D.B.); 2Bruker BioSpin GmbH & Co., Rudolf-Plank-Straße 23, 76275 Ettlingen, Germany; anne.thews@bruker.com (A.T.); roland.leiminger@bruker.com (R.L.)

**Keywords:** COVID-19, SARS-CoV-2, post-acute sequelae of SARS-CoV-2 infection, post-COVID-19 condition, post-COVID-19 syndrome, metabolomics, dyslipidemia, nuclear magnetic resonance

## Abstract

Long COVID, or post-acute sequelae of SARS-CoV-2 infection (PASC), can manifest as long-term symptoms in multiple organ systems, including respiratory, cardiovascular, neurological, and metabolic systems. In patients with severe COVID-19, immune dysregulation is significant, and the relationship between metabolic regulation and immune response is of great interest in determining the pathophysiological mechanisms. We aimed to characterize the metabolomic footprint of recovering severe COVID-19 patients at three consecutive timepoints and compare metabolite levels to controls. Our findings add proof of dysregulated amino acid metabolism in the acute phase and dyslipidemia, glycoprotein level alterations, and energy metabolism disturbances in severe COVID-19 patients 3–4 months post-hospitalization.

## 1. Introduction

During the COVID-19 pandemic, millions of people tested positive for the SARS-CoV-2 virus, hundreds of thousands of patients were hospitalized, and in the middle of 2020, the medical and academic community were already noting Long COVID as an emerging issue in rehabilitation [1,2,3]. Long COVID, or post-acute sequelae of SARS-CoV-2 infection (PASC), is a heterogeneous entity with possible long-term symptoms in multiple organ systems, including respiratory, cardiovascular, neurological, and metabolic systems, among others [4,5,6,7]. Zang and colleagues aimed to characterize PASC using Electronic Health Records (EHR) by comparing ~58 thousand patient data records to ~503 thousand control EHR data records. They found a significantly increased risk for new-onset diagnoses in COVID-19 patients 12 months after the acute disease phase when compared to non-COVID-19 patients, including myopathy, dementia, cognitive problems, skin symptoms, pulmonary fibrosis, dyspnea, pulmonary embolism, thromboembolism, diabetes, cystitis, malaise and fatigue, dizziness, joint pain, and others [4].

Studies worldwide have estimated a high prevalence of long-lasting complications after the COVID-19 acute phase [6,7,8]. A systematic review that included 194 studies (735,006 participants) assessed that 45% of COVID-19 survivors, regardless of hospitalization status, were experiencing unresolved complications for ~4 months [6]. The high prevalence and unclear disease mechanisms lead to situations where healthcare systems are ill-equipped to deal with these patients without clear evidence-based rehabilitation and therapeutic guidelines for PASC patients.

In patients with severe COVID-19, immune dysregulation is significant, and the relationship between metabolic regulation and immune response is of great interest in determining the pathophysiological mechanisms of PASC [9,10]. Quantitative omics approaches, including metabolomics, have been used to understand metabolite profile differences between COVID-19 severity groups and controls, but most of them do not evaluate mid-term and long-term metabolic changes [11,12,13]. The dysregulation of pathways related to energy production and amino acid metabolism has been observed in several studies [11,13,14]. Changes in blood metabolite levels reflect more complex systemic disturbances induced by SARS-CoV-2, which affect the liver, kidneys, and other tissues. Several studies have presented evidence of plausible metabolic reprogramming of the urea cycle and/or the TCA cycle [11,13,15]. COVID-19 is associated with the development of cardiovascular diseases, including dyslipidemia and, specifically, high-density lipoprotein (HDL) dysregulation [16,17]. Another confirmation of disturbed energy metabolism is reports of increased risk of incident diabetes and incident antihyperglycemic use [4,18].

Considering the cardiometabolic nature of PASC, we aimed to characterize the metabolomic footprint of recovering severe COVID-19 patients at three consecutive timepoints and compare the metabolite levels to controls. NMR-based metabolomics was used to investigate metabolic changes during recovery from severe COVID-19. The blood plasma from hospitalized COVID-19 patients was collected at three timepoints: acute phase, a month later, and three to four months later to study metabolite profiles during recovery from severe COVID-19. To see how these profiles change compared to non-COVID-19 samples, we also measured metabolite profiles for population controls (not healthy individuals because of the relatively high prevalence of comorbidities in COVID-19 cases). A Bruker AVNEO 600 MHz IVDr NMR-Solution was employed for metabolomics in the present study. The system utilizes ^1^H NMR spectroscopy, which allows us to quantify not only the free metabolites across different chemical classes but also a variety of lipids, lipoproteins, and their subclasses with exceptionally high reproducibility [19,20,21]. In fact, there are several published studies that have used NMR metabolomics to study the unique metabolic fingerprint of COVID-19 disease in the context of severity, differences among sexes, and virus variants, elucidating the interaction between host metabolism and immune response [14,22,23,24,25]. Our study is designed to address these questions: How do metabolite profiles change during recovery from severe COVID-19 (linear regression for time series data)? Can we identify significantly changed metabolites for patients with and without PASC (time series + phenotype linear regression)? What metabolites significantly differ between COVID-19 patients and controls at each timepoint (*t*-test, fold-change analysis)? What are the significant pathways involved in recovery from severe COVID-19?

## 2. Results

The study included 41 hospitalized patients with severe COVID-19 who were sampled at three timepoints: Timepoint A (1st or 2nd day of hospitalization), Timepoint B (4–5 weeks later), and Timepoint C (3–4 months post-hospitalization). The COVID-19 patients were also compared to 41 population controls (the study design is visualized in Figure 1A). The characteristics of the study cohort are summarized in Table 1. As anticipated, the clinical blood tests showed abnormal results in the acute phase of COVID-19 patients (Timepoint A), corresponding to the previously described state of severe patients in several meta-analyses [26,27]. The systemic response to the infection in our study cohort was affirmed by the high variation observed in platelet, neutrophil, lymphocyte, monocyte counts, alanine aminotransferase, aspartate aminotransferase, gamma-glutamyl transferase, bilirubin, lactate dehydrogenase, creatinine, and C-reactive protein clinical measurements, which indicate renal and hepatic dysfunction and inflammation [26,27].

Primarily, we wanted to identify how metabolite profiles for COVID-19 patients change over time during the recovery process after severe disease and hospitalization. We performed limma linear regression for time series data, using *Subject* as a covariate to adjust for repeated within-subject samples, identifying 115 significant metabolites (*p*-value < 0.05) (Appendix A) across all contrasts (Timepoint C as the reference group); the top 50 are visualized in the Figure 1C heatmap. As expected, the largest number (114) of significantly different (*p*-value < 0.05) metabolites was between the acute phase (Timepoint A) and the latest recovery phase (Timepoint C); similarly, between the acute phase (Timepoint A) and the recovery phase (Timepoint B), 104 significant metabolites (*p*-value < 0.05) were identified. The severe acute phase of COVID-19 patients can explain the significant difference in these contrasts. The top 10 most significantly changed metabolites between the acute and recovery phases (Timepoints A and B, respectively) were GlycB, Glyc, GlycA, Glyc/SPC, threonine, pyruvic acid, Apo-A1, TPA1, H3FC, and HDA1. Interestingly, between the two recovery phases, Timepoints B and C, we identified 17 statistically significant metabolites—the top 10 were methionine, histidine, GlycB, acetic acid, pyruvic acid, Glyc, citric acid, GlycA, threonine, and V4TH—indicating that the recovery process is still ongoing a month (35 ±  7.65 days) after hospitalization. See the Figure 1B venn diagram showing the significant metabolite count across three contrasts. Also, Hotelling’s T^2^ test was performed to detect changes or differences in the multivariate mean of the data over time and to determine if the overall pattern or distribution of these variables changed significantly across the studied timepoints (Appendix A). The top 3 metabolites (GlycA, Glyc, and GlycB by Hotelling’s T^2^ value) are visualized in Figure 1D.

We recognize that our two subgroups, Long COVID and Recovered, each consist of relatively small sample sizes (31 and 10 patients (×3 timepoints), respectively); the division in groups was based on Electronic Health Record data (see methods). However, the strength of our investigation lies in its longitudinal nature, which offers unique insights into the trajectory of COVID-19 patients over time. Metabolome data in blood samples collected at three distinct timepoints allows us to track the disease’s progression and recovery in patients after the acute phase. Therefore, we used the COVID-19 subgroups to test whether there is a significant difference between the Long COVID (*n* = 31) and Recovered (*n* = 10) groups. We performed limma linear regression analysis (Metaboanalyst [18]) with three experimental factors (phenotype, subject, and time) and compared the phenotype groups with time and subject as covariates. When phenotype groups were compared, 27 metabolites were identified as significant (*p*-adjusted < 0.05) (Appendix A), and 19 of those were lipoprotein parameters. The top 10 most significant by adjusted *p*-value are H2CH, H3FC, H2A2, H3A2, choline, H2PL, H3CH, succinic acid, acetone, H3A1, and glucose.

Next, we set out to identify the significantly different metabolites between COVID-19 patients and population controls in each of the timepoints (A, B, and C) and determine whether we see consistent changes across all contrasts that are involved in the pathogenesis of post-acute sequelae of SARS-CoV-2 infection (PASC) or the recovery from acute COVID-19 and identify the significant pathways. When comparing the **COVID-19** acute phase (Timepoint A) vs. the population controls, we performed a *t*-test and identified 116 statistically significant metabolites (FDR < 0.05) (Appendix A) and calculated the fold change (FC) (Appendix A). A total of 44 metabolites were significant with an FC > 1.5. The summary of both analyses is visualized in the volcano plot (Figure 2A), resulting in 13 strongly significant metabolites with a fold change threshold of 2 and FDR < 0.05, 3 of them downregulated (acetic acid, methionine, and proline) and 10 upregulated (threonine, pyruvic acid, V5FC, ornithine, V5CH, acetoacetic acid, 2-Oxoglutaric acid, sacrosine, ethanol, and 2-Hydroxybutyric acid). Figure 2B shows pathway significance (obtained by Global Test pathway enrichment analysis) and pathway impact values (from pathway topology analysis) (Appendix A). Pathway analysis revealed 32 significantly enriched pathways (FDR  <  0.05), including the top 10 by impact values: phenylalanine, tyrosine, and tryptophan biosynthesis; synthesis and degradation of ketone bodies; D-glutamine and D-glutamate metabolism; glycine, serine, and threonine metabolism; alanine, aspartate, and glutamate metabolism; phenylalanine metabolism; arginine and proline metabolism; pyruvate metabolism; glycerolipid metabolism; and citrate cycle (TCA cycle). We also identified some of these pathways as significant in the acute phase vs. controls in our previous targeted metabolomics experiment (LC-MS) [12]. Using liquid chromatography-mass spectrometry (LC-MS) in blood sera, we identified tryptophan (tryptophan, kynurenine, and 3-hydroxy-DL-kynurenine) and arginine (citrulline and ornithine) metabolism, glutamine depletion, and altered metabolism of several amino acids as contributing pathways in the immune response to SARS-CoV-2 in severe COVID-19 [12].

In the present study, we explored how acute phase (Timepoint A) metabolites correlate with hematological and biochemical analyses performed in a clinical lab in the recovery phases (Timepoint B and Timepoint C). Interestingly, the highest correlations (by R values) in Timepoint A metabolites vs. Timepoint B blood tests were ornithine and AST (0.85), proline and alanine aminotransferase ALT (0.83), and asparagine and C-reactive protein (CRP) (0.72). Interestingly, the highest significant correlations in Timepoint A metabolites vs. Timepoint C blood tests were tyrosine and D-dimer (-0.98), glycerol and ALT (0.83), and ornithine and D-dimer (0.79). Figure 2 shows Timepoint A metabolite correlations with blood analysis results at Timepoints B (2C) and C (2D). The correlation coefficients for the significant comparisons can be found in Appendix A.

When comparing the COVID-19 recovery phase/month post-hospitalization (Timepoint B) vs. the population controls, a *t*-test identified 21 statistically significant metabolites (FDR < 0.05) (Appendix A) and calculated the fold change (FC) (Appendix A). A total of 44 metabolites had an FC > 1.5. The summary of both analyses is visualized in the volcano plot (Figure 3A). Acetic acid, V5FC, V5CH, V5PL, and sacrosine were significantly different (FDR < 0.05) in the analyzed comparison with a fold change of ±1.5. Figure 3B shows pathway significance (obtained by Global Test pathway enrichment analysis) and pathway impact values (from pathway topology analysis) (Appendix A). Pathway analysis revealed three significantly enriched pathways (FDR  <  0.05): beta-alanine metabolism, glycolysis/gluconeogenesis, and pyruvate metabolism. Figure 3C visualizes the results of correlation analysis (Appendix A); all statistically significant R values present color between the pair. We identified three correlations as the strongest (by R-value): succinic acid and erythrocytes (0.89), succinic acid and GFR (0.89), and succinic acid and hemoglobin (0.86).

The same analysis strategy was performed in the COVID-19 recovery phase 3–4 months post-hospitalization (Timepoint C) vs. the population controls comparison, summarized in Figure 4A–C. A *t*-test identified 10 statistically significant metabolites (FDR < 0.05) (Appendix A), and 4 metabolites had an FC > 1.5 (Appendix A). The volcano plot (Figure 4A) shows two metabolites (V5FC and ornithine) as significantly different (FDR < 0.05, fold change ±1.5) in the analyzed contrast. Interestingly, pathway analysis (Figure 4B) in this contrast revealed 16 significantly enriched pathways (FDR  <  0.05) compared to only 3 in Timepoint B vs. population controls. In the acute phase (Timepoint C) vs. the population controls, the top three pathways by impact values (among the statistically significant) are: D-glutamine and D-glutamate metabolism; alanine, aspartate, and glutamate metabolism; and glycine, serine, and threonine metabolism (Appendix A). The Figure 4C heatmap visualizes the results of the correlation analysis between Timepoint C metabolites and Timepoint C blood tests; all statistically significant R values present color between the pair. We identified these three correlations as the strongest (by R-value): cholesterol—cholesterol (0.91), triglycerides—triglycerides (TG) (0.89), and HDL-cholesterol—HDL-cholesterol (0.86), which are the same molecules only measured with different methods and labs. The next strongest R values identified were: Apo-B100—cholesterol (0.83), isoleucine—D-dimer (−0,81), and Apo-B100—non-HDL-cholesterol (0.80) (Appendix A).

## 3. Discussion

In this study involving 41 hospitalized COVID-19 patients (measuring metabolites at three timepoints) and 41 population controls, we found proof of dysregulated amino acid metabolism, dyslipidemia, glycoprotein level alterations, and energy metabolism disturbances that we discuss below. Given the high heterogeneity of acute COVID-19 and PASC, the main limitation of our study is the relatively small sample size. However, the application of a longitudinal study design provides greater statistical power and minimizes potential interference with variable complexity at individual levels.

### 3.1. Dysregulations in Amino Acid Metabolism

In our previous metabolomic investigation, we focused on the patients in the acute phase of COVID-19, uncovering significant alterations in amino acid metabolism. Using liquid chromatography-mass spectrometry (LC-MS) on blood sera, we identified tryptophan (tryptophan, kynurenine, and 3-hydroxy-DL-kynurenine) and arginine (citrulline and ornithine) metabolism and glutamine depletion as contributing pathways in the acute phase immune response to SARS-CoV-2 in severe COVID-19. We also reported the altered metabolism of several amino acids (alanine, leucine, histidine, tyrosine, methionine, phenylalanine, asparagine, glutamine, and others) [12]. The current study used NMR on a different cohort of COVID-19 patients, extending the observation period and including samples from the timepoint of 3–4 months post-hospitalization. Limma linear regression identified histidine, methionine, phenylalanine, threonine, glutamine, tyrosine, and others as significant for COVID-19 patient time-series metabolic profiles, but choline and valine were significantly different between the Long COVID and Recovered subgroups. The amino acid concentrations tend to normalize to similar levels as controls for most patients 3–4 months post-hospitalization. The extensive dysregulation of amino acids during the acute phase indicates significant immune system activation, as recently, they have been proposed to be immunometabolites and immunotrasmitters and mark immune cell activity pathways [9,10]. For example, glutamine is important as respiratory fuel for macrophage function, survival, and proliferation of T and B cells [10,28,29,30]. Adding to that, Holmes et al. observed a persistently reduced glutamine/glutamate ratio in most non-hospitalized COVID-19 patients in their study three months post-infection, emphasizing the importance of the relationship between these amino acids in astrocytes in the central nervous system [14]. Interestingly, ornithine, an amino acid involved in the urea cycle, was found to be significantly changed in the present study in time series data linear regression analysis, and we observed that it is higher in the COVID-19 acute phase. For most patients, it lowers to normal levels during the first month of recovery, but for some, it is still significantly higher than levels in controls 3–4 months after hospitalization. We also noted that ornithine significantly positively correlated with alanine aminotransferase and aspartate aminotransferase in the 3–4 month post-hospitalization recovery phase. We discuss liver involvement further in the next paragraphs.

### 3.2. Dyslipidemia in COVID-19

Linear regression analysis for time series data during recovery from severe COVID-19 identified ppolipoprotein-A1 (ApoA-1) and high-density lipoprotein-cholesterol (HDL-Chol) as significantly altered, and when compared to population controls, the Apo-A1 and HDL-Chol levels in blood plasma were still lower a month after hospitalization, and for several patients, they also remained low 3–4 months after hospitalization. Importantly, HDL-Chol was among the significant metabolites when linear regression was used to investigate differences among the Long COVID and Recovered patient groups. This observation is consistent with other studies, which have associated low HDL-Chol levels with poor outcomes [14,31,32]. HDL-Chol is considered “good” cholesterol because it enhances endothelial function and promotes endothelial cell integrity; Apo-A1 forms the largest part of HDL cholesterol. Adding to that, it has been previously proven that the activation of the inflammatory cascade can cause a decrease in HDL-Chol and changes in apolipoprotein profiles, which manifest as impairment in the reverse cholesterol transport mechanism by which the body removes excess cholesterol from peripheral tissues and delivers them to the liver, consequently leading to an increased accumulation of cholesterol in cells [16,33,34]. To note, we also observed a positive correlation between acute phase ApoA-1 and HDL-Chol levels measured by NMR and the recovery phase (1 month and 3–4 months post-hospitalization) HDL-cholesterol levels measured in the clinical lab, indicating that patients are exposed to an abnormal level of lipoproteins for a prolonged time, and it could contribute to endothelial damage observed in COVID-19 patients [35]. We also identified changes in other lipoprotein and triglyceride fractions, indicating dyslipidemia that has been observed in COVID-19 patients before, increasing the risk of cardiovascular complications for patients recovering from COVID-19 [31,32]. Similar results with **^1^**H NMR metabolomics were obtained by Ghini et al. in an Italian COVID-19 patient cohort; they showed that patients with PASC tend to have altered lipoproteomes [25]. Adding to that, in a recent study, dysregulated lipoprotein profiles were noted as more characteristic for non-hospitalized post-acute COVID-19 patients who were experiencing persistent symptoms than those who were considered fully recovered [14]. Our study provides further evidence that dyslipidemia is characteristic of recovery from COVID-19 in hospitalized patients; taken together with the other literature evidence, it is advisable to monitor blood lipid levels for hospitalized COVID-19 patients and consider lipid-lowering medication to avoid poor cardiovascular outcomes.

### 3.3. Glycoprotein Level Alterations

GlycA originates from a subset of glycan N-acetylglucosamine residues, GlycB (branched-chain N-acetyl signals), and Glyc (N-acetylneuraminic acid) on enzymatically glycosylated acute-phase proteins (levels rise or fall in response to inflammation, modification carried out by glycosidases and glycosyltransferases) [36,37,38,39]. Inflammation-induced alterations of N-linked acute-phase glycoproteins result primarily from the addition or removal of sialic acid, galactose, or fucose residues [36,39]. Otvos et al. showed that GlycA can be a robust systematic inflammation marker; Duprez et al. identified it as a marker for inflammation and a strong predictor of cardiovascular events [38,40]. In the present study, GlycA, GlycB, and Glyc significantly changed during recovery from severe COVID-19, as identified by linear regression analysis (the highest levels are in the acute phase). Additionally, GlycA, GlycB, and Glyc showed the highest Hotteling’s T^2^ values across time series data in COVID-19 recovery, indicating the most constant change pattern across patients. When compared to population controls, GlycA, Glyc, and GlycB were still elevated one month after hospitalization, and for most patients, they also stayed elevated 3–4 months after hospitalization when compared to population controls. Importantly, GlycA is a novel marker for chronic, long-term inflammation. Leveraging population-based omics data in a 10-year-long study, increased GlycA levels were found to be chronic within individuals for up to a decade, increasing the risk of severe respiratory infections, which can lead to septicemia and pneumonia [36]. Elevated GlycA positively corresponded to inflammatory cytokines and increased neutrophil activity, suggesting chronic inflammation [36]. We also note that GlycA, GlycB, and Glyc significantly correlated with clinical blood test parameters in our cohort (plasma glucose, LDH (lactate dehydrogenase), GGT (gamma-glutamyl transferase), CRP (C-reactive protein), HDL and LDL cholesterol, neutrophil count, and triglycerides). The chronic inflammation should be studied in large, prospective, recovered COVID-19 and Long COVID patient cohorts to determine the best anti-inflammation therapeutic strategies for patients with PASC.

### 3.4. Energy Metabolism Disturbances

Several studies have presented evidence of disturbed energy metabolism by SARS-CoV-2 virus infection, and our results add evidence in support of these proposed mechanisms as important in recovery after hospitalization [11,41,42]. In our study, two important overlapping pathways of energy metabolism, glycolysis/gluconeogenesis and pyruvate metabolism, were identified as significantly different between COVID-19 patients (at all timepoints) and population controls. Adding to that, the TCA cycle was identified as significantly different between COVID-19 patients in the acute phase and population controls and between COVID-19 patients in the latest recovery phase (3–4 months after hospitalization) and controls. This indicates serious alterations in energy metabolism. Caterino et al. compared metabolomes and cytokine profiles among different clinically severe patients and controls [11]. They proposed a strong connection between the hypoxemia state and the subsequent oxidative stress, which may have a major effect on the mitochondrial energy metabolism and detoxification processes in the hepatocytes, noting that the hepatic urea cycle is the main metabolic pathway involved in the detoxification processes [11]. The urea cycle metabolizes ammonia to urea with a fumarate shunt connecting the urea and TCA cycles within the liver [43]. In a hypoxic state, there is a burden of glucose production (gluconeogenesis) and urea elimination (the urea cycle) in the liver. A recently published study of 39 post-COVID-19 patients in rehabilitation also found high levels of succinic and fumaric acids (TCA cycle metabolites) and associated these observations with hypoxia and inflammation [44]. Although analyzing drug metabolites was out of this study’s scope, a notable fact is that corticosteroids, especially dexamethasone, were used for the treatment of severe COVID-19, and corticosteroids increase hepatic gluconeogenesis, reduce peripheral use of glucose, and increase insulin levels, contributing to hyperglycemia in patients. Acetic acid, lactic acid, pyruvic acid, citric acid, and succinic acid are involved in energy metabolism pathways, and we found them to be significantly changed in COVID-19 patients (linear regression for time series data). The optimistic result in our relatively small cohort study is that 3–4 months post-hospitalization, the levels of these metabolites normalized and were similar to population controls.

Liver tissue is not the only one that changes rapidly in an oxygen-deficient environment caused by pneumonia and lung damage. Marchuenda-Egea and Narro-Serrano described how energy metabolism changes in muscles, liver, and adipose tissue, which leads to the COVID-19 signature blood metabolome in severe COVID-19 patients [41]. Hypoxia caused by pneumonia impacts muscle tissue significantly because oxygen is crucial for the efficient functioning of mitochondria, where it acts as the final electron acceptor in oxidative phosphorylation (the main cellular ATP production process) [41,45]. The muscle tissue adapts to this stress by breaking down muscle proteins to generate energy; one possible pathway for that is the glucose–alanine cycle, which allows myocytes to obtain energy continuously from muscle proteins by oxidizing amino acids [41,46]. The glucose–alanine cycle generates alanine that can be transported to the liver, where it can donate amino groups to pyruvate (to be used in gluconeogenesis) or to oxaloacetate, producing glutamate and initiating the urea cycle [46]. In our study, pathway analysis revealed several significantly different alanine metabolism pathways between COVID-19 patients during recovery phases and population controls, but alanine itself was not significantly altered. Adding to that, during hypoxia, the liver obtains its energy from the oxidation of lipids through β-oxidation, generating ketone bodies [47]. We also measured ketone bodies, and 3-hydroxybutyric acid was higher for some COVID-19 patients at hospitalization but was the same level as controls in both recovery phases. Acetoacetic acid was high for most patients in the acute phase and for some patients a month later, but for all patients, it returned to normal levels 3–4 months post-acute phase. It is interesting to note that Ghini et al. also observed a significant increase in ketone bodies (3-hydroxybutyrate, acetone, and acetoacetate) during the acute phase in most severe patients (they compared mild, severe, fatal, and reference groups), concluding that the higher increase in ketone bodies was attributed to a higher risk of fatal events [24]. An increase in ketone bodies and plasma lipoproteins during the acute phase indicates a greater mobilization of energy sources from adipose tissue, and this contributes to the disturbed lipid metabolism discussed previously.

Finally, our metabolomics study measured glucose, which was identified as statistically significant by the limma linear regression analysis for both the COVID-19 time series data and the Long COVID/Recovered subgroups. The highest levels of glucose were measured during the acute phase. For most patients, glucose tends to go back to normal after 3–4 months of recovery (same levels as controls), but for several patients, it stays higher. It should be noted that one patient had new-onset diabetes and another had increased blood glucose listed in their EHR data during the 12-month post-acute period. In the scientific literature, there is an increasing amount of evidence that some patients develop diabetes as post-acute sequelae of COVID-19 and need glucose-lowering therapies [4,18,48,49]. The literature and our results show that attention to possible hyperglycemia in recovering COVID-19 patients (blood tests) is advisable and should be adequately treated to avoid the development of hyperglycemia-induced complications. We also note that this study had an important limitation that should be addressed in the future: the need for deeper clinical phenotype data on COVID-19 patients in the acute and recovery phases. The availability of these data would provide the possibility to correlate the measured metabolites with characteristic symptoms and complications of acute COVID-19 and PASC.

## 4. Materials and Methods

### 4.1. Study Design

Our study cohort consists of 41 hospitalized severe COVID-19 patients (5 of whom were in the intensive care unit), from whom blood was collected at three timepoints: (1) acute phase (Timepoint A) samples were collected on the 1st or 2nd day of hospitalization; (2) recovery phase (Timepoint B) samples were collected 35  ±  7.65 days later; and (3) later recovery phase (Timepoint C) samples were collected 99  ±  16.79 days after the first sample. The acute phase (Timepoint A) sample collection at the hospital started on 25 May 2020 and was finished by 28 January 2021; all patients had confirmed SARS-CoV-2 infection (with an antibody or qPCR test) at the time of admission to the hospital. The patient recruitment was organized in collaboration with Riga East University Hospital and Vidzeme Hospital (Latvia). During hospitalization, most patients in the cohort received bromhexine hydrochloride or ambroxol hydrochloride as bronchiolitis therapy, dexamethasone (glucocorticoid) as anti-inflammatory therapy, enoxaparin (most often), fraxiparine, or warfarin as an anticoagulant, paracetamol, ascorbic acid, calcium gluconate, sodium chloride, and therapies for individual comorbidities. The collection of longitudinal samples was organized in an ambulatory setting according to the study design. None of the patients were vaccinated with any of the COVID-19 vaccines before hospitalization or during the sample collection period. Hematological and biochemical analyses in the acute phase were performed at the hospital’s clinical lab but during recovery in a certified clinical laboratory (E. Gulbja Laboratorija, Ltd., Riga, Latvia). Health registry data were obtained for the patients, and they were divided into two subgroups (Recovered (*n* = 10) and Long COVID (*n* = 31)) by these criteria: if they had a new post-acute sequelae of SARS-CoV-2 infection (PASC) diagnosis in the 12 months following the COVID-19 acute phase (1-month acute phase and 12-month period as a post-acute period). We used the PASC diagnosis selection from the recently published work of Zang et al. [4]. We also compared COVID-19 patients to population controls (*n* = 41) without acute respiratory infection symptoms during enrollment and 3 months before sample collection (self-reported). All samples were collected during the same period (2020–2021). The study design is visualized in Figure 1A, and cohort characteristics are summarized in Table 1. Written informed consent was obtained from every participant before their inclusion in the study, and the study protocol was approved by the Central Medical Ethics Committee of Latvia (No. 01-29.1.2/928).

### 4.2. Sample Preparation and Instrumental Analysis

The blood samples were collected in EDTA blood collection tubes and centrifuged to separate blood plasma. They were then aliquoted, frozen, and stored according to standard operating procedures (SOPs) of the Genome Database of the Latvian Population (National Biobank [40]). Sample preparation for analysis: 300 μL of thawed blood plasma and 300 μL of Bruker plasma/serum buffer were shaken to mix, then 600 μL was transferred in a 5 mm SampleJet tube used for instrumental analysis. The raw NMR spectrum in samples was recorded using the Bruker IVDrAvance III HD 600 MHz NMR system that uses a temperature-controlled autosampler SampleJet™ and TXI probes resonating with 1H, 13C, and 15N (Bruker BioSpin GmbH, Ettlingen, Germany) [50]. The Bruker IVDrAvance III HD 600 MHz NMR system employs 1H NOESY (Nuclear Overhauser Effect Spectroscopy), JRES (J-RESolved spectroscopy), CPMG (Carr–Purcell–Meiboom–Gill), and PGPE (Pulsed Gradient Perfect Echo) techniques. Together, three modules were applied for fully automated annotation and quantification: B.I. QUANT-PS™ (for small molecules), B.I. LISA™ (for lipoprotein subclasses and subfractions), and B.I. BioBank QC™ (for sample quality control). Sample preparation and instrumental analysis were performed according to SOPs developed by Bruker; it has been shown that by using these standardized SOPs, NMR metabolomics can provide highly reproducible data [20,51,52].

### 4.3. Statistical Analysis

For statistical analysis of metabolites, we used a merged table of all quantified metabolites, including free metabolites (no protein denaturing performed) across different chemical classes (alcohols and derivatives, amines and derivatives, amino acids and derivatives, carboxylic acids, essential nutrients, keto acids and derivatives, sugars and derivatives, sulfones, and technical additives), N-acetylated- glycoproteins (GlycA and GlycB), Glyc (N-acetylneuraminic acid), SPC (supramolecular phospholipids composite), quantified lipids and lipoproteins (triglycerides, cholesterol, LDL-chol, HDL-chol, LDL-phos, HDL-phos, Apo-A1, Apo-B100), and concentrations of lipoprotein VLDL, IDL, LDL, and HDL classes and subclasses. We also included biologically relevant proportion calculations: Apo-B100/Apo-A1, Glyc/SPC, and LDL-cholesterol (LDCH)/HDL-cholesterol (HDCH) as possible biomarkers across contrasts. Together, we analyzed 169 quantified features in 164 samples. See the quantified input features for all samples in Appendix A.

For statistical analysis, we used web-based MetaboAnalyst software 5.0 [53]. To normalize metabolite data, we applied log transformation (base 10) and Pareto scaling (mean-centered and divided by the square root of the standard deviation of each variable).

### 4.4. Linear Regression for Time Series Data Analysis in COVID-19 Patients

The underlying method is based on limma [54]. We analyzed two models: (1) time series data and (2) time series + phenotype (Recovered/Long COVID). For model (1), the reference group was set to be Timepoint C when compared to Timepoints A and B, but Timepoint B was used as a reference group to compare with Timepoint A; for model (2), the direction of comparison was Long COVID vs. Recovered. In both analyses, *Subject,* indicating one patient, was used as a covariate to adjust for repeated within-subject samples. For time-pattern analysis, we used multivariate empirical Bayes statistical time-series analysis (MEBA) at Metaboanalyst, a method to rank the features by calculating Hotelling’s T^2^. It can be applied to determine if these variables’ overall pattern or distribution changes significantly across timepoints.

### 4.5. Univariate Analysis for Two-Group Comparisons

To identify the significantly different metabolites between COVID-19 patients and population controls at each of the timepoints (A, B, and C) and determine whether we see consistent changes across all contrasts that are involved in the pathogenesis of or recovery from acute COVID-19, we also performed the univariate analysis in each of the interesting contrasts with Metaboanalyst: *t*-test to determine significance and fold change analysis calculation to determine the level of changed metabolite concentration. To visualize the significance and fold change analysis results in the volcano plot, −log_10_(*p*-value) and log_2_FoldChange were calculated in Metaboanalyst and visualized with Prism Graphpad 9.

### 4.6. Pathway Analysis

We also used Metaboanalyst to determine significant pathways in each of the contrasts; for that, a set of features was used (only quantified metabolites that could be annotated). Log transformation and Pareto scaling were applied for data normalization. The reference pathway library was selected for *Homo sapiens (KEGG).* The enrichment method (Global Test) was used to calculate adjusted *p*-values and −log10(*p*-value) for the plot. The reference pathway library was selected for *Homo sapiens (KEGG).*

### 4.7. Biochemical Analysis and Metabolite Correlations

We also analyzed the correlations of the metabolites with the hematological and biochemical markers from laboratory analyses at different timepoints. In each pairwise correlation, we excluded any patient if the respective laboratory analysis was not performed or the metabolite/marker was below detection level, and the correlation coefficient was only calculated if data were available for at least five patients. The calculation was performed using the non-parametric Kendall’s tau.

## 5. Conclusions

While currently there are no regulator-approved therapies that directly target the root causes of Long COVID symptoms, and the causes themselves are not widely understood, metabolomics data from longitudinal studies of recovering patients can identify the molecules behind observed complications to advise possible risks in recovery from severe COVID-19. From our data and the literature evidence, we recommend checking recovering COVID-19 patients for hyperglycemia and dyslipidemia with widely available markers in the blood and considering chronic inflammation as a potential risk of developing complications over time.

## Figures and Tables

**Figure 1 ijms-25-01523-f001:**
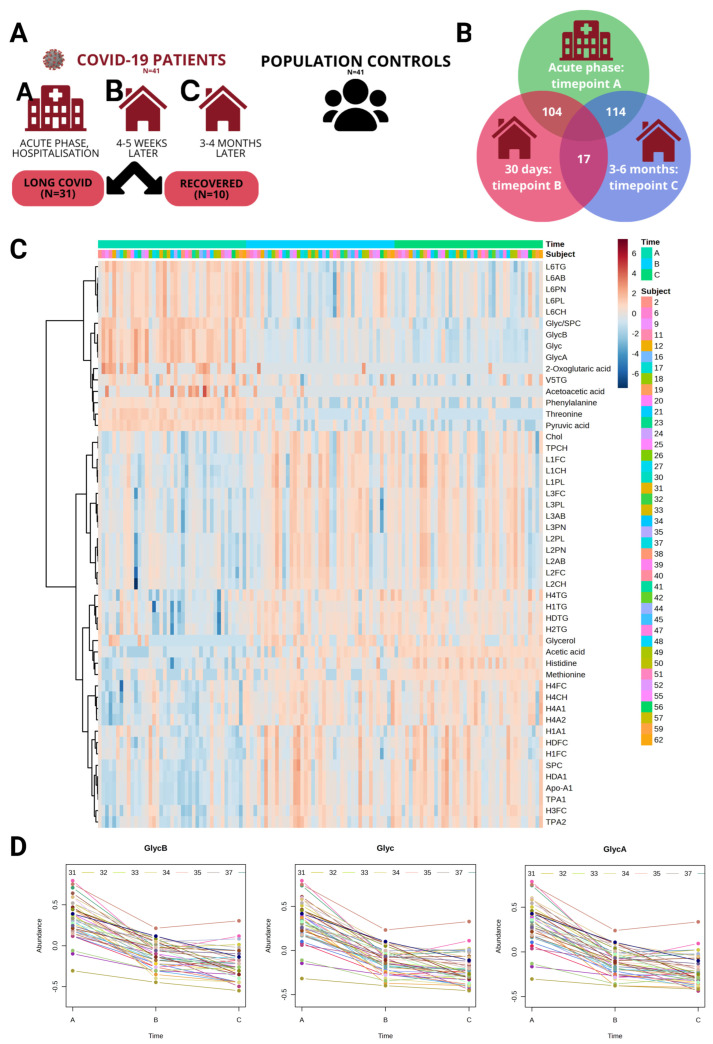
(**A**) Study design. The study cohort consists of 41 hospitalized COVID-19 patients, for whom samples were collected at three timepoints: (1) acute phase (Timepoint A) on the 1st or 2nd day of hospitalization; (2) recovery phase (Timepoint B) 35  ±  14.7.65 days later; and (3) later recovery phase (Timepoint C) 99  ±  16.79 days after the first sample. (**B**) Venn diagram showing the count of statistically significant metabolites between timepoints A, B, and C. (**C**) Top 50 significantly changed features (metabolites) based on limma linear regression analysis for time series data visualized in the heatmap; each row conforms to a specific metabolite expressed in a normalized, log-transformed concentration value, each column represents one sample, and the samples are arranged by timepoint. Distance measure: euclidean; clustering algorithm: ward. (**D**) Time-course profiles for the top 3 features (by Hotelling’s T^2^ value) from multivariate empirical Bayes statistical time-series analysis (MEBA); each line represents one patient.

**Figure 2 ijms-25-01523-f002:**
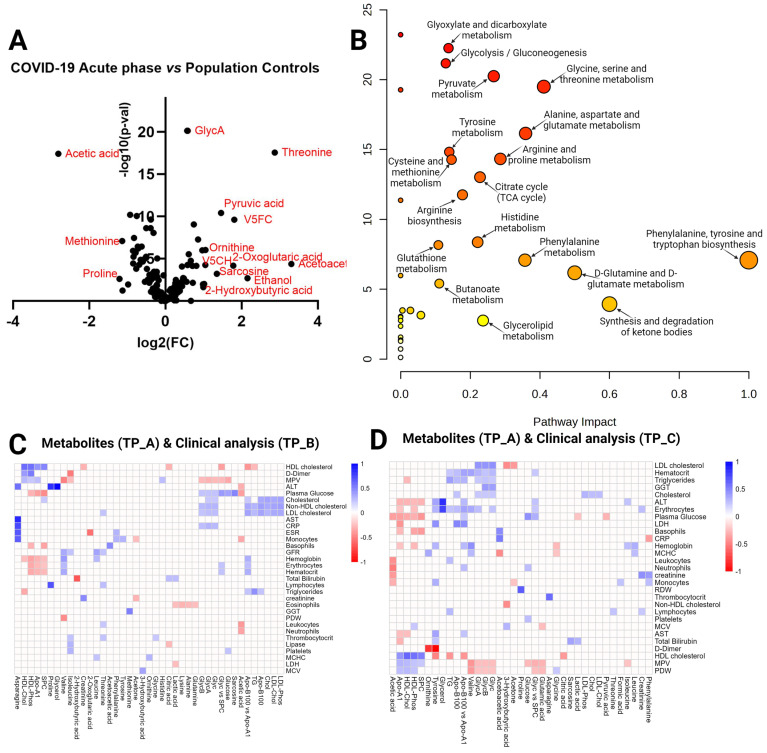
Timepoint A vs. population controls. (**A**) Volcano plot visualizing the distribution of metabolites in the analyzed contrast, significance (−log_10_(*p*-value)) versus log_2_ fold change is plotted on the *y* and *x* axes, respectively, resulting in 13 strongly significant metabolites (fold change > 1.5 and FDR < 0.05) indicated in red in the plot. (**B**) Scatterplot representing the most relevant metabolic pathways from the KEGG library, arranged by adjusted *p* values (obtained by Global Test pathway enrichment analysis) on the *y*-axis and pathway impact values (from pathway topology analysis) on the *x*-axis. The node color is based on its *p* value, and the node radius is determined based on pathway impact values. (**C**) Timepoint A metabolites (*x* axis) vs. Timepoint B blood tests (*y* axis) in COVID-19 patients. Heatmap with the most clinically relevant metabolite (Timepoint A) correlations with biochemical and hematological analysis results from the clinical lab at recovery phase Timepoint B (~month after admission at the hospital), color represents the strength of the relationship and its direct or inverse nature; only the pairs with sufficient data and significant correlations (*p*-value > 0.05) are shown. (**D**) Timepoint A metabolites (*x* axis) vs. Timepoint C blood tests (*y* axis). Heatmap visualizing metabolite (acute phase) correlations with biochemical and hematological analysis results in recovery phase Timepoint C (~month after admission at the hospital), color represents the strength of the relationship and its direct or inverse nature; only the pairs with sufficient data and significant correlations (*p*-value > 0.05) are shown.

**Figure 3 ijms-25-01523-f003:**
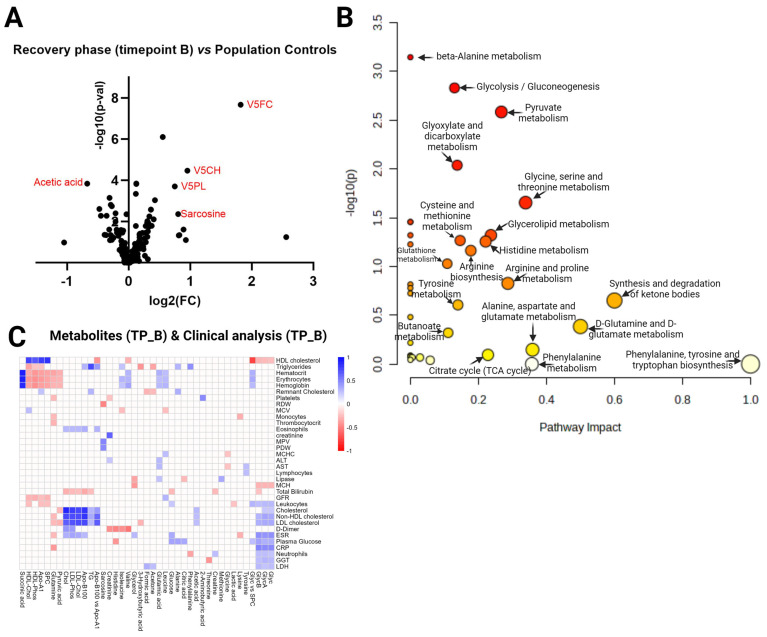
Timepoint B vs. population controls. (**A**) Volcano plot visualizing the distribution of metabolites in the analyzed contrast, significance (−log_10_(*p*-value)) versus log_2_ fold change is plotted on the *y* and *x* axes, respectively. (**B**) Scatterplot representing the most relevant metabolic pathways from the KEGG library, arranged by adjusted *p* values (obtained by Global Test pathway enrichment analysis) on the *y*-axis and pathway impact values (from pathway topology analysis) on the *x*-axis. The node color is based on its *p* value, and the node radius is determined based on pathway impact values. (**C**) Timepoint B metabolites (*x* axis) vs. Timepoint B blood tests (*y* axis) in COVID-19 patients. Heatmap visualizing statistically significant metabolite correlations with biochemical and hematological analysis results in recovery phase Timepoint B (~month after admission at the hospital), color represents the strength of the relationship and its direct or inverse nature; only the pairs with sufficient data and significant correlations (*p*-value > 0.05) are shown.

**Figure 4 ijms-25-01523-f004:**
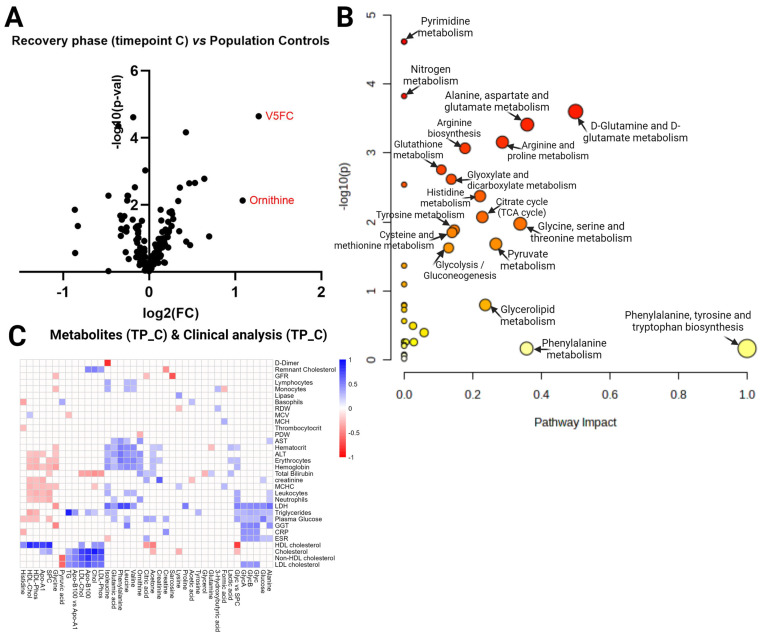
Timepoint C vs. population controls. (**A**) Volcano plot visualizing the distribution of metabolites in the analyzed contrast, significance (−log_10_(*p*-value)) versus log_2_ fold change is plotted on the *y* and *x* axes, respectively. (**B**) Scatterplot representing the most relevant metabolic pathways from the KEGG library, arranged by adjusted *p* values (obtained by Global Test pathway enrichment analysis) on the *y*-axis and pathway impact values (from pathway topology analysis) on the *x*-axis. The node color is based on its *p* value, and the node radius is determined based on pathway impact values. (**C**) Timepoint C metabolites (*x* axis) vs. Timepoint C blood tests (*y* axis) in COVID-19 patients. Heatmap visualizing statistically significant metabolite correlations with biochemical and hematological analysis results in recovery phase Timepoint B (~month after admission at the hospital), color represents the strength of the relationship and its direct or inverse nature; only the pairs with sufficient data and significant correlations (*p*-value > 0.05) are shown.

**Table 1 ijms-25-01523-t001:** Characteristics of the study participants.

Variable, Mean (SD) or n (%)	COVID-19 Patients		Population Controls
Male/Female (n, %)	17 (41.46%)/24 (58.54%)		17 (41.46%)/24 (58.54%)
Age, average ± SD (years ± SD)	56.63 ± 13.16		53.10 ± 11.41
BMI, average (kg/m^2^ ± SD)	34.40 ± 22.95		27.08 ± 4.60
Time in hospital (days ± SD)	9.18 ± 3.25		-
Smoker/non-smoker (n, %)	3 (7.32%)/38 (92.68%)		3 (7.32%)/38 (92.68%)
	**Comorbidities ***		
Number of patients with comorbidities (yes/no)	30 (73.17%)/11 (26.83%)		23 (56.10%)/18 (43.90%)
Hypertension (n, %)	17 (41.46%)		9 (21.95%)
Type 2 Diabetes Mellitus (n, %)	3 (7.32%)		3 (7.32%)
Other cardiovascular disease (n, %)	10 (24.39%)		13 (31.71%)
Oncological (n, %)	2 (4.88%)		3 (7.32%)
**Clinical measurements ****			
**Average**	**Acute COVID-19**	**Recovery phase** **(1 month)**	**Recovery phase** **(3–4 months)**
Leukocytes (μL, SD)	5.71 (2.15)	6.19 (1.37)	5.67 (1.67)
Hemoglobin (g/dL, SD)	13.24 (1.45)	13.85 (1.28)	14.32 (1.43)
Hematocrit (%, SD)	39.91 (4.07)	41.47 (3.08)	41.26 (8.05)
Platelets (μL, SD)	173.13 (99.07)	267.10 (49.20)	238.17 (59.95)
Neutrophils (μL, SD)	2.41 (1.49)	3.21 (1.01)	3.00 (1.15)
Lymphocytes(μL, SD)	0.58 (0.54)	2.02 (0.66)	1.97 (0.66)
Monocytes (μL, SD)	0.24 (0.22)	0.59 (0.22)	0.50 (0.15)
Eosinophils (μL, SD)	0.03 (0.05)	3.03 (1.70)	3.05 (1.62)
ALT (U/l, SD)	23.82 (18.49)	39.23 (28.53)	30.95 (17.54)
AST (U/l, SD)	28.75 (13.74)	28.45 (13.68)	26.75 (12.91)
GGT (U/l, SD)	78.33 (99.33)	44.43 (43.10)	28.31 (30.13)
Bilirubin (μmol/L, SD)	6.64 (3.22)	13.30 (5.31)	11.26 (4.60)
LDH (U/L, SD)	295.00 (168.87)	215.70 (38.10)	189.67 (71.51)
Creatinine (μmol/L, SD)	72.45 (19.87)	68.65 (11.53)	70.57 (18.49)
CRP (mg/L, SD)	35.05 (42.05)	3.69 (2.80)	3.39 (4.28)
D-dimer (mg/mL, SD)	0.60 (0.12)	0.48 (0.31)	0.25 (0.15)

* Comorbidities for population controls were self-reported. ** Clinical measurements in the acute COVID-19 phase were performed at an in-house hospital clinical laboratory, but the recovery phase measurements were performed in the largest local network of certified clinical laboratories outside hospitals. SD: standard deviation; BMI: body mass index; ALT: alanine aminotransferase; AST: aspartate aminotransferase; GGT: gamma-glutamyl transferase; LDH: lactate dehydrogenase; CRP: C-reactive protein.

## Data Availability

The data presented in this study are available upon request from the corresponding authors due to privacy or ethical restrictions.

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
