# Peer review of "Longitudinal NMR-Based Metabolomics Study Reveals How Hospitalized COVID-19 Patients Recover: Evidence of Dyslipidemia and Energy Metabolism Dysregulation"

_ijms, 2024, doi:10.3390/ijms25031523_

Round 1

Reviewer 1 Report

Comments and Suggestions for Authors

There is a rich literature on the use of metabolomics to identify the profile of covid-19 in the serum or plasma of patients in the acute and post-acute phase, both through MS and NMR. Especially regarding the latter, many studies have been conducted as part of a covd-network promoted by the company Bruker BioSpin. Some co-authors of this manuscript belong to Bruker but surprisingly ignore almost all references involving members of that consortium (in particular: Nicholson & Holmes; Luchinat & Turano; Millet; Guenther; Spraul). In reality there are many points of contact between the works produced by the members of the consortium and the results found in the present study. It therefore becomes essential that the introduction and discussion take this into account. As also admitted by the authors, this study suffers from the limitation of a relatively small number of subjects. However, results in common with previous studies could serve to attribute greater statistical validity to the present findings. The pathway analysis published here could be compared with the metabolic model recently proposed in PLOS Pathogens https://doi.org/10.1371/journal.ppat.1011787.

The results presented in the manuscript under consideration should be presented also in the light of possible differences among SARS-CoV-2 variants. From the reported description it is not possible to understand when samples were collected and whether differences among variants could represent a confounding factor. It is not even possible to judge whether the PASC profile can be influenced by vaccination status.

Reviewer 2 Report

Comments and Suggestions for Authors

The manuscript is devoted to the study of the serum metabolic profile of acute and post COVID-19 patients. The main idea to search for some statistically significant metabolites in the same patients during 3 time points was interesting and perspective. Also, the idea to compare COVID-19 patients with population controls with very similar characteristics was also very reasonable. However, some important data should be added into the main text of the manuscript.

1) The manuscript should contain more information about patients and their state:

1.1 What were the inclusion/exclusion criteria for patients?

1.2 The severity of the acute-COVID-19 should be mentioned for all patients including the information about fibrosis on chest CT (CT 1–4).

1.3 The frequency of different patients’ complaints in the stage of post-COVID-19 (fatigue, cognitive problems and so on) should be well-structured here to understand the heterogeneity of the patients.

1.4 The changes in the clinical state of patients during 2 and 3 time points should be presented. It is unlikely that patients experienced the same set of symptoms throughout the entire recovery period.

1.5 This leads to the question: how did the observed changes in the metabolite profile correlate not only with biochemical parameters, but also with clinical manifestations of the post-COVID-19?

1.6 Table 1 should contain reference values for clinical measurements to understand how many variables were out of normal values in different stages.

1.7 The deviation for the second time point is too much and strange: 35 +-14.7.65. (line 395)

2) The manuscript should contain more information about the NMR method used:

2.1 How many times was the same sample analyzed?

2.2 What was the RSD of the metabolite signals?

2.3 Was the RSD of the analysis taken into account when conducting the statistical analysis between COVID-19 patients and controls?

2.4 The discussion contains the information on the use of LC-MS method (line 249). What were the reasons to use NMR instead of LC-MS in the current study?

3) According to ICO-10-CM code the “long Covid” is called “post COVID-19 condition” and I suppose that authors should include this and some other related names like “post COVID-19 syndrome” in keywords.

4) The Figure 1 C and D should be placed into the Suppls as they are too small to see the details. Also, the Figure 1 D does not demonstrate any differences between 2 and 3 point, however, there were mentioned in the text that the differences were statistically significant.

5) Also, some other metabolic studies in post COVID-19 could be interesting for the authors to explain and correlate their results:

Holmes, E.; Wist, J.; Masuda, R.; Lodge, S.; Nitschke, P.; Kimhofer, T.; Loo, R.L.; Begum, S.; Boughton, B.; Yang, R.; et al. Incomplete Systemic Recovery and Metabolic Phenoreversion in Post-Acute-Phase Nonhospitalized COVID-19 Patients: Implications for Assessment of Post-Acute COVID-19 Syndrome. J. Proteome Res. 2021, 20, 3315–3329.

Ghini, V.; Meoni, G.; Pelagatti, L.; Celli, T.; Veneziani, F.; Petrucci, F.; Vannucchi, V.; Bertini, L.; Luchinat, C.; Landini, G.; et al. Profiling metabolites and lipoproteins in COMETA, an Italian cohort of COVID-19 patients. PLoS Pathog. 2022, 18, e1010443.

Sorokina, E.; Pautova, A.; Fatuev, O.; Zakharchenko, V.; Onufrievich, A.; Grechko, A.; Beloborodova, N.; Chernevskaya, E. Promising Markers of Inflammatory and Gut Dysbiosis in Patients with Post-COVID-19 Syndrome. J. Pers. Med. 2023, 13, 971.

Reviewer 3 Report

Comments and Suggestions for Authors

COVID-19 is an important medical and social problem, so a better understanding of its pathogenesis and clinical characteristics is an important challenge.

Comments:

1. Clinical characterization of the patients is not sufficient: what was the severity of COVID-19 in the patients: was there pneumonia (volume) and respiratory failure? What treatment did all patients receive?

2. Were these unvaccinated patients? Were the patients in the control group immunized? Did the patients in the control group have previous COVID-19 disease? How was this assessed? Were antibodies assessed prior to inclusion?

3. It is recommended that the inclusion and exclusion criteria be described in more detail. What were the clinical criteria for time points? What were the clinical criteria for Acute phase, Recovery phase and later Recovery phase? Hospitalization could have occurred on any day of illness depending on the characteristics of the course. Therefore, the day of hospitalization cannot characterize the acuteness of the process.

4. The abstract is recommended to be structured

5. A few minor comments: make references to the literature [1-3] instead of [1]-[3]. Table 1 says BML, but probably meant BMI.

6. In Table 1, clarify the hemoglobin and neutrophil levels in each group. It is recommended to add in a note what the numbers and figures in brackets in the Clinical measurements section mean.

Reviewer 4 Report

Comments and Suggestions for Authors

This paper aims to investigate the effect of Covid on immune response using NMR-based metabolomics to check on the immune dysregulation. While this study is interesting that it checks on the metabolic changes during covid and post-covid, the study could be more interesting if it considers if the patients suffer from mild or severe symptoms, and if the patients have been vaccinated. The present study is based on the assumption that everyone have the same symptoms and hence the variations for results within the group itself maybe big. It may require a larger sample size to make the finding in this study more convincing. 

While for the write-up, avoid using first person view of writing in scientific writing. Please amend accordingly. 

For the selection of respondents, the age group is about 50+, please justify. Is it because this group is more likely (higher risk) to be associated to other diseases? 

The SD in table 1 is very high for certain variables/ measurement, some are even much higher than the average values, is the data still considered valid? It indicates a huge variation and may imply the need for a bigger sample size.

line 134, did the author mean threshold?

Some parts in discussion are rather general, would be good to associate it to certain disease with the potential markers identified. 

Comments on the Quality of English Language

Some minor syntax errors to be corrected. Please check through the entire manuscript. 

Round 2

Reviewer 1 Report

Comments and Suggestions for Authors

I'm satisfied with the revision besides a final minor point: it seems to me that the authors have not included the references suggested by Reviewer 2 (at point 5). That request shall be satisfied before publication.

Author Response

Thank you for the suggestion; we supplemented the discussion of our results with the suggested sources; see lines 293-297, 328-333, and 382-385.

Reviewer 3 Report

Comments and Suggestions for Authors

Although the authors answered some of the questions in the response letter, data for many of these questions were not added to the article itself. It is recommended to improve the clinical description of patients in the article and add the missing data to the study limitations. It is also recommended to add information on which hospital (country) the study was conducted. 

Author Response

Thank you for the note. We supplemented the study cohort description, including hospital and country (see lines 455-466), and added the missing data as a study limitation in discussion, lines 434-438.

Round 3

Reviewer 3 Report

Comments and Suggestions for Authors

The authors added some necessary data, including the limitations of the study